# Public interest in palliative care in Latin America: A Google Trends analysis

**Brayan Miranda-Chavez** [1,2‡], **Niels Pacheco-Barrios** [3,4‡], **Alvaro Taype-Rondan** [4,5] *

1 Universidad Privada de Tacna, Tacna, Perú, 2 Centro de Estudios e Investigación en Educación Médica y Bioética, Universidad Privada de Tacna, Tacna, Perú, 3 Alberto Hurtado Medical School, Cayetano Heredia Peruvian University, Lima, Perú, 4 EviSalud–Evidencias en Salud, Lima, Perú, 5 Universidad San Ignacio de Loyola, Unidad de Investigación para la Generación y Síntesis de Evidencias en Salud, Lima, Perú

‡ BMC and NPB authors share first authorship on this work.
* alvaro.taype.r@gmail.com

## Abstract

The demand for palliative care is increasing globally, yet a notable lack of awareness continues to present a significant obstacle to its widespread adoption. The use of digital tools like Google Trends can help gauging public interest in specific topics. We used Google Trends to conduct a systematic search of terms related to palliative care from January 1, 2010, to May 10, 2023. The results were filtered by location, including worldwide and Latin American countries. We found a global increase in searches for terms related to palliative care, with a peak in December 2022 associated with the death of Brazilian footballer Pelé. Countries like Brazil, Mexico, and Colombia mirrored this trend, while others like Argentina and Peru did not. Interest in palliative care is on the rise in Latin America, albeit with notable regional variations.

## Introduction

The rise in non-communicable diseases and enhanced life expectancy worldwide have escalated the demand for palliative care [1]. Consequently, many countries acknowledge its significance for a comprehensive and effective healthcare system [2].

In several low- and middle-income countries, palliative care is still in process of development. significant challenge faced in these regions is the limited awareness and knowledge surrounding palliative care. According to the Global Atlas of Palliative Care's 2017 report, out of the 56.8 million recorded deaths, only 25.7 million individuals were able to access palliative care services. To put this into perspective, in Latin America alone, an estimated 2.5 million people could potentially benefit from palliative care, yet a mere 1% of them actually have access [3].

Recently, the monitoring and analysis of public interest in health topics through digital tools have gained traction. Google Trends, a notable tool, which offers valuable insights into the search trends of specific keywords over time and location. Utilizing Google Trends in public health research can track the efficacy of interventions, monitor disease outbreaks, and investigate social trends.

From 2005 to 2019, two studies in the United States indicated an upward trend in searches for "palliative care" and "hospice care" [2, 4]. Similarly, a European study from 2004 to 2020

**Data Availability Statement:** The data has been taken from Google Trends, a platform that is available to the public. Additionally, two Excel documents were added: Supplementary Material 02, which includes the raw data from the analysis,

and Supplementary Material 03, which contains the data used for figure creation.

**Funding:** The author(s) received no specific funding for this work.

**Competing interests:** The authors have declared that no competing interests exist.

observed a rise in interest in "palliative care" and "advance healthcare directives" [5]. However, there is a dearth of analogous studies focused in Latin American countries.

In this context, we performed the following study to explore the level of interest in palliative care in Latin America through the analysis of Google Trends data from 2010 to 2023.

## Methodology

The search was conducted through Google Trends (http://www.google.com/trends), a service offered by Google, which furnishes a Relative Search Volume (RSV) value for specific keywords within the desired time frame.

The search was conducted during May 2023, and the data extraction encompassed the time span from January 1, 2010, to May 10, 2023. The included search terms were: [paliativos+paliativo+paliativa+paliativas], [palliative], [eutanasia], [euthanasia], [muerte digna], [death with dignity], [sedación terminal], [terminal sedation], [quimioterapia paliativa], and [palliative chemotherapy]. The term [paliativos+paliativo+paliativa+paliativas] was filtered by region, including worldwide, Brazil, Mexico, Colombia, Argentina, Peru, Venezuela, Chile, Guatemala, Ecuador, and Bolivia. These 10 Latin-American countries represent the most populated countries of the regions [6].

All the Google Trends RSV raw data was obtained and exported into Microsoft Excel. The same program was used to create the line charts to visualize tendencies. When we discovered a peak in any of the tendencies, we conducted a Google News search to identify media event that may have caused such peak. In addition, we compared the mean RSV (from 2019 to 2021) per country and the palliative care level of development using a scatterplot, using the software Tableau.

The level of palliative care development published by David Clark [7] was based on data collected through an online questionnaire, completed by experts from each country, including the 193 United Nations Member States, two observer states, as well as Kosovo, Somaliland, and Taiwan, China, all in the year 2017. Furthermore, 10 key indicators were established and categorized into 6 distinct groups: No known palliative care activity, Capacity-building in palliative care, Isolated palliative care provision, Generalized palliative care provision, Palliative care at the preliminary stage of integration, and Palliative care at an advanced stage of integration.

For data analysis, a linear regression was performed to evaluate the slopes of the trend for [paliativos+paliativo+paliativa+paliativas]. Finally, an interrupted time series was performed using the RStudio Software, to estimate differences in RSV tendencies before and after the announcement of use of palliative care by the soccer player Pelé. This was modeled as a coefficient change (slope change) in the trend.

It is important to note that the methods used for data collection and analysis adhered to the terms and conditions set by the data source.

## Results

A global peak is evident in December 2022 in the worldwide trend graph for the term [paliativos+paliativo+paliativa+paliativas] (Fig 1A). A similar trend is seen in the data for Brazil, Mexico, and Colombia (Fig 1B). However, in the case of Argentina, Peru, Venezuela, Chile, Guatemala, Ecuador, and Bolivia, there is no clear rising trend (Fig 1C and 1D). Additionally, an upward trend is noted for the term [palliative] when evaluated worldwide (Fig 1E).

The interrupted time series analysis of the terms [paliativos+paliativo+paliativa+paliativas] worldwide showed that there was no significant change in the trend's slope before and after this peak. (Fig 2)

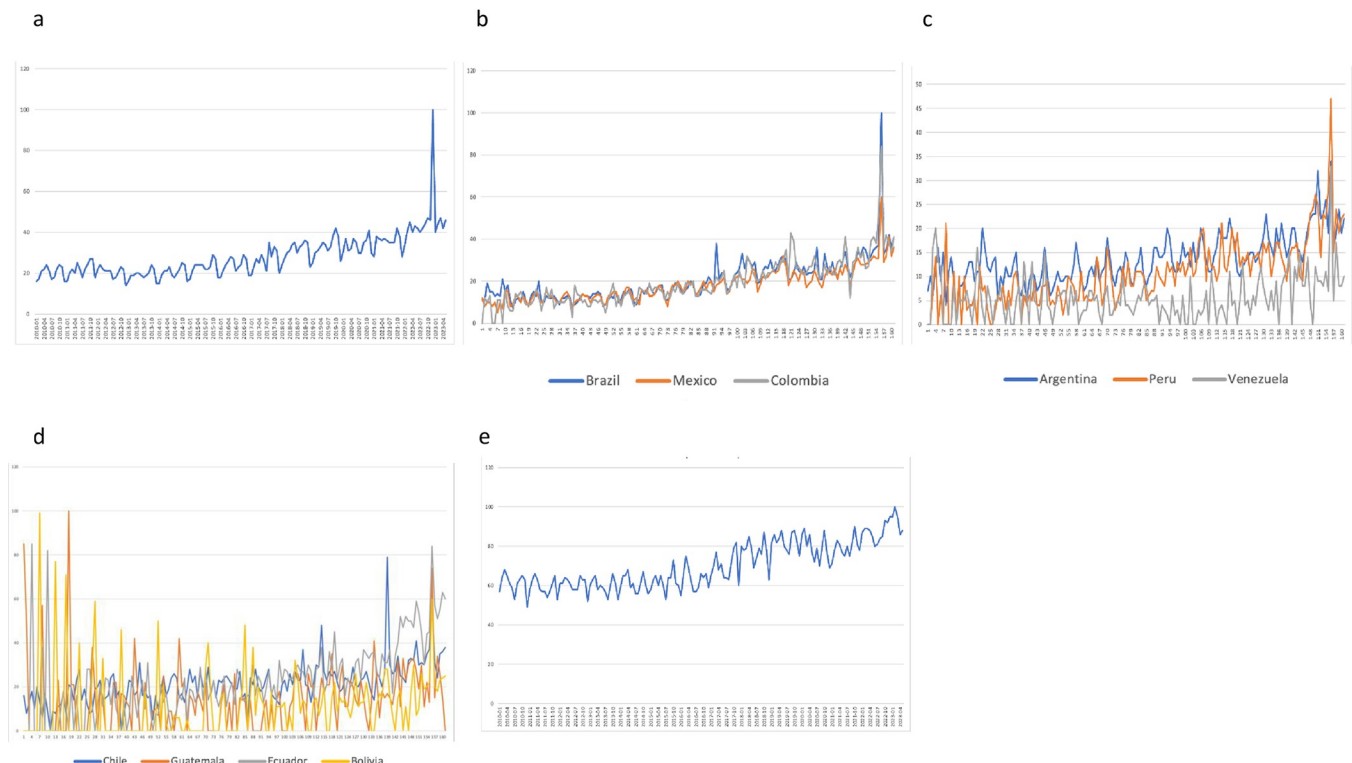

**Fig 1. Trends for the terms [paliativos+paliativo+paliativa+paliativas].** a) Worldwide b) Brazil, Mexico, Colombia c) Argentina, Peru, Venezuela d) Chile, Guatemala, Ecuador, Bolivia e) Worldwide trend for the term "palliative".

In the comparison of trends for palliative care-related terms in English and Spanish, we observed that the terms [muerte digna] and [Death with dignity] have unrelated peaks in the RSV, while the other terms showed no observable trends. (Fig 3) Other terms such as [salud] and [health], and [paliativismo] in Portuguese also showed a positive trend (S1 and S2 Figs).

Our findings reveal a correlation between countries with the most advanced palliative care development and a correspondingly elevated average RSV for palliative care-related terms (Table 1). Nevertheless, it's worth noting that Ecuador stands out as an outlier in this context. Despite having a low level of palliative care development, it demonstrates a notably high mean RSV.

## Discussion

We found a rising interest in palliative care, evident in both Spanish and English languages. In Latin America, this increase has been notorious in specific countries. Accordingly, two studies [2, 4] utilizing Google Trends showed a consistent increase in interest in palliative care in the United States, while a study conducted in three European countries between 2004 and 2020 revealed a similar trend for terms such as "euthanasia," "palliative care," and "advance health care directives" [5]. This increase coincided with the implementation of laws, policies, and services promoting palliative care.

This growing interest may be attributed partly to the expansion of palliative care programs worldwide [8]. Enhanced media coverage and heightened engagement on social media platforms like Twitter [9], Facebook [10], and TikTok [11] also potentially contribute to this trend.

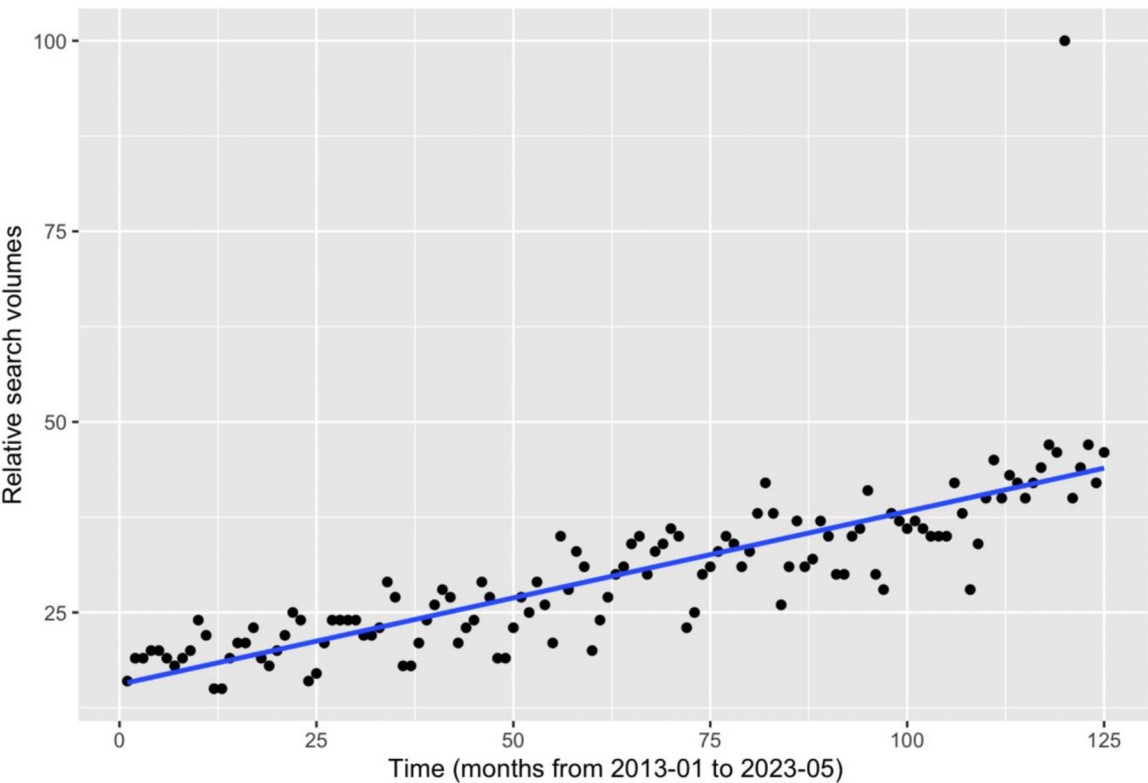

**Fig 2. Interrupted time analysis series of the terms [paliativos+paliativo+paliativa+paliativas] worldwide.**

Our results show that, in Latin America, most countries exhibit an upward trend in interest in palliative care. However, Bolivia and Guatemala do not display a distinct trend, likely due to minimal search activity. This increasing interest contrasts with the region's scarce palliative care services. The Latin American Atlas of Palliative Care [12] reports that, in 2017, there were only 1,562 such services in the region, insufficient compared to the actual needs.

Interestingly, we found that countries with more advanced palliative care provisions showed increased interest, although not statistically significant, possibly due to limited sample sizes. A study [4] analyzing U.S. states found a positive correlation between the availability of palliative care programs and search volume, suggesting a potential association between program availability and public interest and awareness. The rising interest may also bolster the development of palliative care programs or, alternatively, the two factors may reciprocally reinforce each other.

We found a notable spike in searches for palliative care in December 2022. This aligns with the transfer of Pelé, the renowned former Brazilian football player, to a palliative care unit on December 3rd, 2022 [13], an event that garnered significant attention, particularly in Latin America where football is highly popular.

This phenomenon mirrors a study utilizing Google Trends, which found a surge in searches for "comfort care" in the United States in 2018, coinciding with the announcement that former First Lady Barbara Bush would forgo further medical treatment in favor of "comfort care" [14].

Following December 2022, the Relative Search Volumes (RSV) reverted to November levels before resuming an upward trend. This indicates that Pelé high-profile case led to a transient increase in interest, akin to prior instances observed in Google Trends, such as Ebola-related search spikes. However, some events might maintain elevated interest levels over an extended period [15].

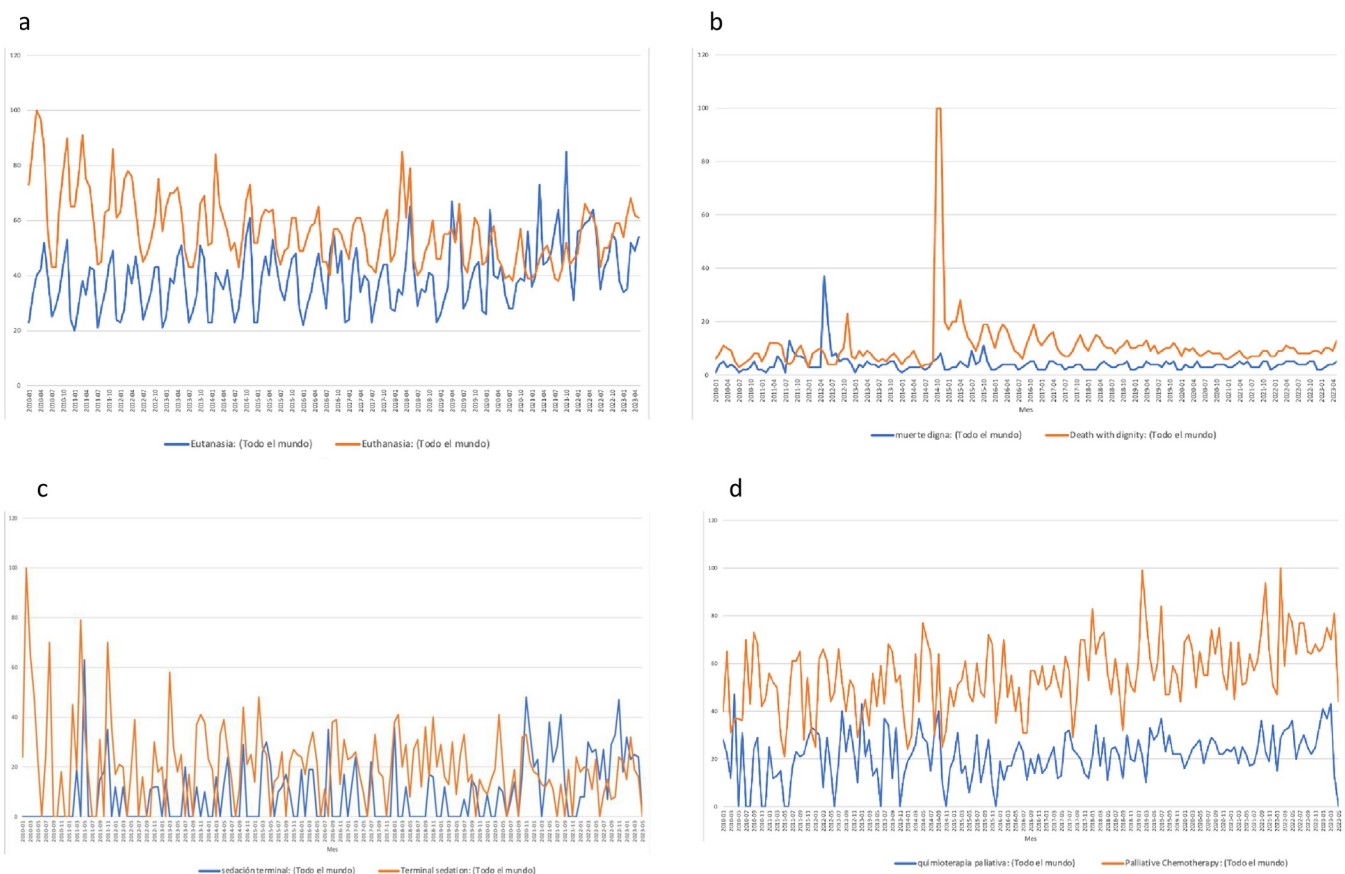

**Fig 3. Trends comparing palliative care related terms in English and Spanish.** a) Eutanasia / Euthanasia b) Muerte digna / Death with dignity c) sedación terminal /Terminal sedation. d) quimioterapia paliativa /Palliative Chemotherapy.

The evaluated terms, such as "[eutanasia]", "[euthanasia]", "[muerte digna]", "[death with dignity]", "[sedación terminal]", "[terminal sedation]", "[quimioterapia paliativa]", and "[palliative chemotherapy]", did not exhibit a clear trend, potentially due to their infrequent usage. It is important to note that while palliative care has traditionally focused on cancer patients, it now encompasses individuals with various serious chronic illnesses. As a result, the terms

**Table 1. Comparison between mean relative search volume and palliative care level of development.**

| Country | Palliative care level of development (assigned numerical value) | Mean relative search volume (RSV) for the 2019–2021 period |
|---|---|---|
| Brazil | Generalize Provision (4) | 26.14 |
| Mexico | Preliminary stage of integration (5) | 22.94 |
| Colombia | Generalized Provision (4) | 26.86 |
| Argentina | Preliminary stage of integration (5) | 16.13 |
| Peru | Isolated provision (3) | 14.02 |
| Venezuela | Isolated provision (3) | 6.14 |
| Chile | Preliminary stage of integration (5) | 25.89 |
| Guatemala | Isolated provision (3) | 17.28 |
| Ecuador | Isolated provision (3) | 29.80 |
| Bolivia | Isolated provision (3) | 12.19 |

"[quimioterapia paliativa]" and "[palliative chemotherapy]" may be less commonly used today, which could explain the low search frequency observed in our study.

Additionally, while the search of the terms "[salud]" and "[health]" showed an increasing trend only following the COVID-19 pandemic in 2019/2020, the terms "[paliativos/paliativo/ paliativa/paliativas]" have shown a continuous rise since the 2010s. This suggests that the growing interest in palliative care is not merely a secondary effect of the increased interest in health due to the COVID-19 pandemic. Furthermore, the COVID-19 pandemic and other public health crises in Latin America, such as the Zika virus outbreak, do not appear to correlate with significant changes in the overall trend of interest in palliative care.

While a positive correlation between the VRS and the level of development of palliative care was observed, this finding remains inconclusive due to the limited number of countries analyzed and notable exceptions. For instance, Ecuador demonstrates a low level of development but a high search volume, whereas Argentina shows a high level of development but a low search volume.

### Limitations

Our study has some limitations. Firstly, Google Trends' transparency is limited by the absence of absolute numbers. Secondly, the platform exclusively provides data on online searches, excluding information from alternative sources or offline behaviors. Thirdly, identified trends could be influenced by factors beyond healthcare policies and legal norms.

Nonetheless, this is the inaugural study evaluating interest in palliative care through Google Trends. The results offer insight into the search dynamics and establish a baseline for future studies and interventions.

### Conclusions

In conclusion, the data from 2013 to 2023 reveals a rising trend in the interest in palliative care worldwide and particularly in Latin America, as evidenced by search terms. Notably, countries with more advanced palliative care infrastructures exhibited heightened interest. Furthermore, an anomalous spike in interest in 2022 is seemingly linked to the high-profile case of Pelé, the renowned former soccer player; while this did not appear to boost the upward trend in palliative care interest.

### Supporting information

**S1 Fig. Térm [health + salud].**
(TIF)

**S2 Fig. Term [palliativism] in the portuguese language.**
(DOCX)

**S1 Data. Data set used for statistical analysis.**
(XLSX)

### Author Contributions

**Conceptualization:** Brayan Miranda-Chavez, Alvaro Taype-Rondan.

**Formal analysis:** Niels Pacheco-Barrios, Alvaro Taype-Rondan.

**Investigation:** Brayan Miranda-Chavez.

**Methodology:** Brayan Miranda-Chavez, Niels Pacheco-Barrios, Alvaro Taype-Rondan.

**Writing – original draft:** Brayan Miranda-Chavez, Niels Pacheco-Barrios.

**Writing – review & editing:** Brayan Miranda-Chavez, Niels Pacheco-Barrios, Alvaro Taype-Rondan.

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
