## [Decision Letter · Decision Letter 0]

23 May 2024

PONE-D-23-43665Public interest in palliative care in Latin America: A Google Trends analysisPLOS ONE

Dear Dr. Taype-Rondan,

Thank you for submitting your manuscript to PLOS ONE. After careful consideration, we feel that it has merit but does not fully meet PLOS ONE’s publication criteria as it currently stands. Therefore, we invite you to submit a revised version of the manuscript that addresses the points raised during the review process.

We look forward to receiving your revised manuscript.

Kind regards,

Martin Schneider

Academic Editor

PLOS ONE

Journal Requirements:

3. In your Methods section, please include additional information about your dataset and ensure that you have included a statement specifying whether the collection and analysis method complied with the terms and conditions for the source of the data.

5. Please include your tables as part of your main manuscript and remove the individual files. Please note that supplementary tables (should remain/ be uploaded) as separate ""supporting information"" files.

Additional Editor Comments:

Dear Dr Taype-Rondan,

Thank you for submitting your brief manuscript “Public interest in palliative care in Latin America: A Google Trends analysis.” It focuses on a topic of certain interest and corroborates previous studies from other geographical areas.

In spite of the positive opinion of the one reviewer, there are several issues and open questions:

How did other health-related keywords change over time? A quick look at “Google Trends” shows that the relative search volumes for both “health” and “salud” have increased from 2010 to 2023. This finding may put your results into a context that you have not discussed.

Palliative care has historically focused on cancer patients. This is not the case any more; patient with manifold severe chronic diseases are provided palliative care. Thus, the search term “palliative chemotherapy” may have become less representative of palliative care internet queries since the late 2010s.

What about searches in other South American languages than Spanish, like Quechua for the Andes, or “paliativismo,” as apparently used in Brazil? Attempts to include ethnic minorities should be visible in the analysis.

During the selected time frame, public health crises occurred in Latin America. Particularly the Zika virus outbreak and the COVID pandemic can be associated with needs for palliative care. How do they show in your analysis?

The correlation in figure 4 may not satisfy all statisticians…

In the references, “disponible en” should be translated into English.

Reviewers' comments:

Reviewer's Responses to Questions

**Comments to the Author**

1. Is the manuscript technically sound, and do the data support the conclusions?

Reviewer #1: Yes

2. Has the statistical analysis been performed appropriately and rigorously? 

Reviewer #1: Yes

3. Have the authors made all data underlying the findings in their manuscript fully available?

Reviewer #1: Yes

4. Is the manuscript presented in an intelligible fashion and written in standard English?

Reviewer #1: Yes

5. Review Comments to the Author

Reviewer #1: Manuscript Review:

Title: Public Interest in Palliative Care in Latin America: A Google Trends Analysis

Manuscript Review:

The manuscript presents a comprehensive analysis of public interest in palliative care in Latin America using Google Trends data. The study addresses the increasing demand for palliative care globally and the challenges associated with limited awareness and access, particularly in low- and middle-income countries. By leveraging Google Trends as a digital tool, the manuscript explores search trends related to palliative care from January 2010 to May 2023, focusing on Latin American countries.

The introduction provides a thorough background on the significance of palliative care in addressing the needs of individuals with life-limiting illnesses and the gaps in awareness and access observed in Latin America. The rationale for utilizing Google Trends to analyze public interest in palliative care is well articulated, emphasizing its utility in tracking social trends and monitoring health-related topics.

In the methodology section, the manuscript details the search process conducted through Google Trends and the selection of relevant search terms. The inclusion of specific keywords related to palliative care, as well as the filtering of data by region, demonstrates a systematic approach to data collection. The manuscript also employs line charts and scatterplots to visualize trends and correlations, enhancing the clarity of the analysis.

The results section presents the main findings of the study, highlighting the rising trend in searches for palliative care worldwide and in Latin American countries. Notable peaks in search volume are identified and associated with specific events, such as the case of Brazilian footballer Pelé, providing valuable insights into factors influencing public interest in palliative care.

The discussion section interprets the findings in the context of existing literature and offers explanations for observed trends, including the correlation between interest in palliative care and the availability of palliative care services. The manuscript also discusses regional variations in search activity and their implications for palliative care development in Latin America, contributing to our understanding of the evolving landscape of palliative care awareness and access.

Overall, the manuscript effectively communicates the study's objectives, methods, and findings, providing valuable insights into public interest in palliative care in Latin America. The integration of Google Trends data with other sources enriches the analysis and enhances our understanding of the challenges and opportunities in advancing palliative care in the region. However, further discussion on the limitations of Google Trends data and potential biases in search behavior could strengthen the manuscript's comprehensiveness.

6. PLOS authors have the option to publish the peer review history of their article (what does this mean?). If published, this will include your full peer review and any attached files.

Reviewer #1: **Yes: **Doaa Attia

---

## [Author Response · Author response to Decision Letter 0]

9 Jun 2024

Dear Editor,

I hereby submit for your consideration the revised version of the manuscript, now titled "Public interest in palliative care in Latin America: A Google Trends analysis," for possible publication in your prestigious journal.

All co-authors have thoroughly reviewed the suggestions, comments, and annotations provided on the previous version of the manuscript, to which we received feedback on May 23rd of this year. We greatly appreciate the time spent on your review and have addressed each suggestion as follows:

1. How did other health-related keywords change over time? A quick look at “Google Trends” shows that the relative search volumes for both “health” and “salud” have increased from 2010 to 2023. This finding may put your results into a context that you have not discussed.

R: The terms [salud] and [health] have consistently shown a high search volume, but it wasn't until the COVID-19 pandemic that they experienced a significant peak, which has been sustained since then. On the other hand, terms related to palliative care have been on a positive trend even before the pandemic. It is worth noting that this is a great contribution, and therefore, it has been deemed appropriate to include supplementary material showing the trend of the terms [salud] and [health] in Google Trends.

2. Palliative care has historically focused on cancer patients. This is not the case any more; patient with manifold severe chronic diseases are provided palliative care. Thus, the search term “palliative chemotherapy” may have become less representative of palliative care internet queries since the late 2010s.

R: This is another valuable contribution; therefore, it has been deemed appropriate to add a paragraph in the discussion. This paragraph is the penultimate one of the discussions and is cited verbatim below: << The evaluated terms, such as "[eutanasia]", "[euthanasia]", "[muerte digna]", "[death with dignity]", "[sedación terminal]", "[terminal sedation]", "[quimioterapia paliativa]", and "[palliative chemotherapy]", did not exhibit a clear trend, potentially due to their infrequent usage. It is important to note that while palliative care has traditionally focused on cancer patients, it now encompasses individuals with various serious chronic illnesses. As a result, the terms "[quimioterapia paliativa]" and "[palliative chemotherapy]" may be less commonly used today, which could explain the low search frequency observed in our study..>>

3. What about searches in other South American languages than Spanish, like Quechua for the Andes, or “paliativismo,” as apparently used in Brazil? Attempts to include ethnic minorities should be visible in the analysis.

 R: This analysis with the Portuguese term [paliativismo] has been added to the supplementary 

 material. Likewise, it's worth noting that a precise term has not been found for the Quechua or Aymara languages.

4. During the selected time frame, public health crises occurred in Latin America. Particularly the Zika virus outbreak and the COVID pandemic can be associated with needs for palliative care. How do they show in your analysis?

R: This positive trend in favor of palliative care does not seem to have been influenced by the public health issues that LATAM has experienced. Accordingly, we have added the following paragraph in the discussion section: “Additionally, while the search of the terms "[salud]" and "[health]" showed an increasing trend only following the COVID-19 pandemic in 2019/2020, the terms "[paliativos/paliativo/paliativa/paliativas]" have shown a continuous rise since the 2010s. This suggests that the growing interest in palliative care is not merely a secondary effect of the increased interest in health due to the COVID-19 pandemic. Furthermore, the COVID-19 pandemic and other public health crises in Latin America, such as the Zika virus outbreak, do not appear to correlate with significant changes in the overall trend of interest in palliative care.”

5. The correlation in figure 4 may not satisfy all statisticians…

R: Although a positive correlation was found between the VRS and level of development, this result is inconclusive due to the small number of countries analyzed and significant exceptions. For instance, Ecuador exhibits a low level of development but a high search volume, while Argentina displays a high level of development but a low search volume. Additionally, this was added as a new paragraph in the Discussion section, which is cited verbatim below: <<While a positive correlation between the VRS and the level of development of palliative care was observed, this finding remains inconclusive due to the limited number of countries analyzed and notable exceptions. For instance, Ecuador demonstrates a low level of development but a high search volume, whereas Argentina shows a high level of development but a low search volume.>>

6. In the references, “disponible en” should be translated into English.

R: This term has been translated in the references section.

---

## [Editor Report · Decision Letter 1]

17 Jun 2024

Public interest in palliative care in Latin America: A Google Trends analysis

PONE-D-23-43665R1

Dear Dr. Taype-Rondan,

We’re pleased to inform you that your manuscript has been judged scientifically suitable for publication and will be formally accepted for publication once it meets all outstanding technical requirements.

Kind regards,

Martin Schneider

Academic Editor

PLOS ONE

Additional Editor Comments (optional):

Dear Dr Taype-Rondan,

Thanks for resubmitting the revised manuscript.

It is almost ready for print, but 3 issues should still be resolved.

Please spell “Pelé” consistently throughout the manuscript.

“disponible en” translates to “available at.”

Please change the correlation in figure 4 to a dotted line or remove it completely.

With best regards

Martin Schneider

---

## [Editor Report · Acceptance letter]

9 Jul 2024

PONE-D-23-43665R1 

PLOS ONE

Dear Dr. Taype-Rondan, 

I'm pleased to inform you that your manuscript has been deemed suitable for publication in PLOS ONE. Congratulations! Your manuscript is now being handed over to our production team.

Kind regards, 

on behalf of

Dr. Martin Schneider 

Academic Editor

PLOS ONE